# Na^+^-Coupled Nutrient Cotransport Induced Luminal Negative Potential and Claudin-15 Play an Important Role in Paracellular Na^+^ Recycling in Mouse Small Intestine

**DOI:** 10.3390/ijms21020376

**Published:** 2020-01-07

**Authors:** Michiko Nakayama, Noriko Ishizuka, Wendy Hempstock, Akira Ikari, Hisayoshi Hayashi

**Affiliations:** 1Laboratory of Physiology, School of Food and Nutritional Sciences, University of Shizuoka, Yada 52-1, Suruga-ku, Shizuoka 422-8526, Japan; m-7ka8ma@outlook.jp (M.N.); n-ishizuka@u-shizuoka-ken.ac.jp (N.I.); w.hempstock@gmail.com (W.H.); 2Laboratory of Biochemistry, Department of Biopharmaceutical Sciences, Gifu Pharmaceutical University, Gifu 501-1196, Japan; ikari@gifu-pu.ac.jp

**Keywords:** tight junction, Na^+^ cotransport, leaky epithelia

## Abstract

Many nutrients are absorbed via Na^+^ cotransport systems, and therefore it is predicted that nutrient absorption mechanisms require a large amount of luminal Na^+^. It is thought that Na^+^ diffuses back into the lumen via paracellular pathways to support Na^+^ cotransport absorption. However, direct experimental evidence in support of this mechanism has not been shown. To elucidate this, we took advantage of claudin-15 deficient (*cldn15^−/−^*) mice, which have been shown to have decreased paracellular Na^+^ permeability. We measured glucose-induced currents (Δ*I*_sc_) under open- and short-circuit conditions and simultaneously measured changes in unidirectional ^22^Na^+^ fluxes (Δ*J*^Na^) in Ussing chambers. Under short-circuit conditions, application of glucose resulted in an increase in Δ*I*_sc_ and unidirectional mucosal to serosal ^22^Na^+^ (∆*J*^Na^_MS_) flux in both wild-type and *cldn15^−/−^* mice. However, under open-circuit conditions, Δ*I*_sc_ was observed but ∆*J*^Na^_MS_ was strongly inhibited in wild-type but not in *cldn15^−/−^* mice. In addition, in the duodenum of mice treated with cholera toxin, paracellular Na^+^ conductance was decreased and glucose-induced ∆*J*^Na^_MS_ increment was observed under open-circuit conditions. We concluded that the Na^+^ which is absorbed by Na^+^-dependent glucose cotransport is recycled back into the lumen via paracellular Na^+^ conductance through claudin-15, which is driven by Na^+^ cotransport induced luminal negativity.

## 1. Introduction

Nutrient absorption in the small intestine is essential for assimilation of nutrients required for energy and growth. To meet these requirements, nutrients are efficiently absorbed from the luminal side, which is the external milieu. In nutrient absorbing cells, many nutrients cross the luminal membrane via specific transporters which require cotransport with luminal Na^+^. Na^+^ coupling allows nutrients to be transported against their concentration gradient from a low luminal concentration to a higher intracellular concentration. The driving force for nutrient transporters is provided by the electrochemical Na^+^ gradient across the luminal membrane. This Na^+^ gradient is produced by the Na-K-ATPase on the basolateral membrane which keeps the intracellular Na^+^ concentration low. Therefore, it is expected that Na^+^ nutrient cotransport concomitantly occurs with net Na^+^ absorption from the luminal (extracellular) to the serosal side [1].

There are many Na^+^ nutrient cotransport systems in the small intestine, such as glucose, amino acids, etc. [2,3,4]. It is, therefore, envisaged that Na^+^-dependent nutrient absorption mechanisms require a large amount of luminal Na^+^, which should be theoretically predictable as follows: For a healthy adult male, daily energy intake is 2500 kcal. The percentage of energy derived from the three major macronutrients in a typical Western diet is carbohydrates (52%), fat (33%), and protein (15%) [5]. Since protein and carbohydrates yield 4 kcal/g and 9 kcal/g for fat, the total amount of daily intake of carbohydrates, fat, and protein is about 325, 92, and 93 g, respectively. The resultant products of digestion produce large amounts of monosaccharides and amino acids in the lumen. The intake of 325 g of carbohydrates will yield approximately 1.8 moles of monosaccharide, which is mostly composed of glucose. The primary transporter that mediates glucose transport in the intestinal brush border membrane has been identified as Na^+^-dependent glucose transporter SGLT1, which operates with a transport stoichiometry of 2 Na^+^:1 glucose [6,7]. Thus, it is estimated that 3.6 moles of Na^+^ are needed for the absorption of glucose only, which corresponds to 23 L of isotonic NaCl solution, containing 210 g NaCl. This volume of fluid is approximately two times larger than that of extracellular fluid, which is 20% of body weight. With regards to protein, assuming that the average molecular weight of an amino acid residue in protein is 110, i.e., 93 g of protein will yield approximately 0.8 moles of amino acids, which needs an equivalent amount of luminal Na^+^ for amino acid absorption from the luminal side. Taking into consideration all of the Na^+^-dependent nutrient absorption, nutrient absorption mechanisms would require a large amount of luminal Na^+^.

How does the small intestine meet the requirements for such a large amount of Na^+^ for absorption of nutrients? One possible source is digestive juices. In humans, it is thought that 9L/day of isotonic fluid enters the lumen of the proximal small intestine through diet and secretion of the upper digestive tract [8]. This fluid is composed of 2 L from diet, 1 L of saliva, 2 L of gastric juice, 1 L of bile, 2 L of pancreatic juice, and 1 L of secretions from the small intestine, which contain only 0.8 moles of Na^+^ in total. Another luminal Na^+^ providing mechanism would be via the paracellular pathway. It is well known that small intestinal epithelia are classified as leaky epithelia, which means its paracellular conductance is ~90% or more of the total tissue conductance and it is cation selective (P_Na_ > P_Cl_) [9]. However, the physiological relevance of cationic selectivity of the paracellular pathway remains to be fully elucidated. The properties of the paracellular pathway are dependent on tight junctions, which occur where epithelial cells are closely connected to each other. The claudin family of tight junction proteins is critical in determining the paracellular ionic permeability and selectivity. The claudin family consists of 27 integral membrane proteins [10]. The subcellular distribution of the claudins varies; while claudin-2 and -15 are exclusively in the tight junctions, other isoforms, e.g., claudin-4 and -7, are found in the basolateral membrane [11,12,13]. This diverging pattern of distribution suggests that individual claudins are engaged in different physiological functions. We previously showed that a loss of claudin-15 decreased the luminal Na^+^ concentration and glucose absorption is inhibited in mouse small intestine [14]. It is hypothesized that Na^+^ diffuses back into the lumen via the paracellular pathway to support nutrient absorption [15]. However, direct experimental evidence in support of this idea has not been shown. In this study, to investigate whether paracellular Na^+^ conductance through claudin-15 is involved in this Na^+^ recycling system and to elucidate the role of Na^+^-nutrient cotransport induced luminal negativity for Na^+^ recycling, we measured glucose-induced short-circuit currents (Δ*I*_sc_) under open- and short-circuit conditions in Ussing chambers. In addition, we simultaneously measured changes in unidirectional mucosal to serosal ^22^Na^+^ (∆*J*^Na^_MS_) flux in wild-type mice and compared them with those of 15 deficient (*cldn15^−/−^*) mice. Furthermore, to understand the mechanism of oral rehydration therapy, which is based on the notion that the Na^+^ that is absorbed by Na^+^-glucose cotransport enters into systemic circulation [16], we measured glucose-induced unidirectional ∆*J*^Na^_MS_ in cholera toxin-diarrhea model mice. The results showed that under the short-circuit conditions in wild-type mice, luminal application of glucose resulted in an increase in ∆*J*^Na^_MS_ which corresponded to the amplitude of Δ*I*_sc_. However, under open-circuit conditions, a Δ*I*_sc_ increase was observed but ∆*J*^Na^_MS_ was strongly inhibited. These results suggest that Na^+^ is recycled back to the lumen under physiological conditions. In *cldn15^−/−^* mice, a robust increase in ∆*J*^Na^_MS_ was observed under open-circuit conditions, suggesting that the efficiency of Na^+^-recycling systems was reduced. This phenomenon was also mimicked by cholera toxin-induced diarrhea in wild-type mice.

Our study demonstrates that the Na^+^ which is absorbed by Na^+^-nutrient cotransport is recycled back into the lumen via paracellular Na^+^ conductance through claudin-15, which is driven by Na^+^ cotransport induced luminal potential.

## 2. Results

### 2.1. Baseline Na^+^ Absorption Mechanisms in Wild-Type Mice

Baseline *I*_sc_ and ^22^Na^+^ unidirectional fluxes were measured simultaneously in the same preparations in wild-type mice (Table 1). 

The unidirectional mucosal to serosal ^22^Na^+^ flux (*J*^Na^_MS_: 51.4 ± 2.3 µmol/cm^2^/h) was larger than the serosal to mucosal flux (*J*^Na^_SM_: 24.6 ± 1.7 µmol/cm^2^/h). This result suggests that Na^+^ absorption occurred in the baseline conditions. The magnitude of the net ^22^Na^+^ flux (*J*^Na^_Net_: 26.9 ± 1.5 µmol/cm^2^/h) was significantly greater than that of the basal short-circuit current (*I*_sc_) (2.4 ± 0.5 µmol/cm^2^/h) suggesting that the net Na^+^ absorption occurs mostly via an electroneutral Na^+^/H^+^ exchange mechanism. To determine the contribution of Na^+^/H^+^ exchanger-3 isoform (NHE3) to baseline *I*_sc_ and ^22^Na^+^ unidirectional fluxes, we used the NHE3 specific inhibitor S3226. In the presence of the S3226, the basal net Na^+^ flux was decreased by 60% (*J*^Na^_Net_: 10.6 ± 3.9 µmol/cm^2^/h). Basal *I*_sc_ and transmural tissue conductance (*G*_t_) were not significantly changed by S3226 (*p* = 0.29 and 0.25, *I*_sc_ and *G*_t_, respectively), suggesting that paracellular ion permeability is not affected by S3226. However, 10 µM S3226 did not completely inhibit basal Na^+^ absorption. Although other NHE isoforms could be involved in basal Na^+^ absorption, these results suggest that the basal net Na^+^ absorption is mainly dependent on NHE3 transport, consistent with what has been previously reported [17].

### 2.2. Activation of SGLT1 Concomitantly Increases Mucosal to Serosal ^22^Na^+^ Fluxes under Short-Circuit Conditions

As shown in Figure 1A, the addition of glucose to the mucosal side increased *I*_sc_ (∆*I*_sc_ 13.7 ± 0.8 µmol/cm^2^/h) in wild-type mice. *J*^Na^_MS_ was also significantly increased (Figure 1B, Δ*J*^Na^_MS_, 12.5 ± 0.8 µmol/cm^2^/h, *n* = 5) after luminal application of glucose, while the unidirectional serosal to mucosal ^22^Na^+^ flux (*J*^Na^_SM_) was not significantly changed after the addition of luminal glucose (Figure 1B, 25.5 ± 1.1 vs. 23.5 ± 1.1 µmol/cm^2^/h, *p* = 0.21, before and after the addition of glucose, respectively).

These results suggest that glucose-induced Δ*J*^Na^_MS_ is mediated by the Na^+^-dependent glucose transporter SGLT1. To confirm this, we conducted four experiments. First, the specific SGLT1 inhibitor phloridzin (0.2 mM) was added to the luminal side, and glucose-induced Δ*J*^Na^_MS_ was measured. In the presence of phloridzin, both glucose-induced ∆*I*_sc_ and Δ*J*^Na^_MS_ increments were totally abolished (*I*_sc_: 1.1 ± 0.3 vs. −0.1 ± 0.3 µmol/cm^2^/h, *p* = 0.06, *J*^Na^_MS_: 48.8 ± 4.2 vs. 46.1 ± 1.9 µmol/cm^2^/h, *p* = 0.88, *n* = 3 before and after addition of glucose, respectively). Second, transepithelial ^36^Cl^−^ unidirectional flux was measured with or without luminal glucose. It is thought that Na^+^-coupled glucose transport from the lumen to intercellular spaces provides an osmotic gradient that results in passive ion movement through tight junctions [18]. However, there was no discernable changes in ^36^Cl^−^ unidirectional fluxes with or without luminal glucose (Δ*J*^Cl^_Net_: 14.5 ± 1.8 vs. 14.0 ± 1.7 µmol/cm^2^/h, *p* = 0.54, *n* = 5 before and after addition of glucose, respectively). We next assessed the contribution of NHE3 to glucose-induced Δ*J*^Na^_MS_ increments, as Na^+^-coupled glucose uptake stimulates NHE3 transport activity in the mouse jejunum [19]. The above-mentioned glucose-induced Δ*J*^Na^_MS_ increments could be mediated by NHE3 and not SGLT1. To examine this possibility, we measured the glucose-induced ^22^Na^+^ unidirectional fluxes in the presence of the NHE3 specific inhibitor S3226 [20]. As shown in Figure 1C, the addition of glucose to the mucosal side resulted in an increase of the *I*_sc_ (∆*I*_sc_: 17.5 ± 1.6 µmol/cm^2^/h, *n* = 5), which is slightly higher in the absence of S3226 (*p* = 0.05). Robust glucose-induced Δ*J*^Na^_MS_ increment was also observed in the presence of S3226 (Figure 1D, ∆*J*^Na^_MS_: 20.2 ± 1.8 µmol/cm^2^/h), suggesting that NHE3 is not responsible for glucose-induced ^22^Na^+^ unidirectional flux increments. Finally, we assessed whether glucose metabolic pathways contribute to Δ*J*^Na^_MS_ increments or not. The non-metabolizable glucose analogue α-methyl-d-glucose (αMDG) was used instead of d-glucose, and the αMDG-induced ^22^Na^+^ unidirectional fluxes were measured. The addition of αMDG to the mucosal side increased the *I*_sc_ in a dose-dependent manner. This change in *I*_sc_ conformed to Michaelis–Menten kinetics, and these values were not significantly different as compared with those of glucose (*V*_max_: 551 ± 47 vs. 661 ± 45 µA/cm^2^, *p* = 0.14, *K*_m_: 10.9 ± 2.9 vs. 5.1 ± 1.2 mM, *p* = 0.14, glucose and αMDG, respectively). As shown in Figure 1E,F, the addition of 10 mM αMDG to the mucosal side increased the *I*_sc_ (∆*I*_sc_: 12.7 ± 1.7 µmol/cm^2^/h) and ∆*J*^Na^_MS_ (12.3 ± 1.6 µmol/cm^2^/h). These values are not significantly different as compared with those of glucose values (*p* = 0.55, 0.89, ∆*I*_sc_ and ∆*J*^Na^_MS_, respectively). Taken together, these results suggest that glucose-induced increment of ∆*J*^Na^_MS_ is mediated by SGLT1.

We next assessed the quantitative relationship between ∆*I*_sc_ and ∆*J*^Na^_Net_ under short-circuit conditions, where ∆*J*^Na^_Net_ is plotted as a function of ∆*I*_sc_ (Figure 2, closed circles). 

We found that there was a positive correlation between these values (r square = 0.62). Taken together, under short-circuit conditions, these results suggest that glucose-induced Δ*J*^Na^_MS_ is mainly mediated by SGLT1 and there is less contribution, if any, from other Na^+^ transport mechanisms involved in glucose-induced Δ*J*^Na^_MS_ increments.

### 2.3. Activation of SGLT1 does not Increase Mucosal to Serosal ^22^Na^+^ Fluxes under Open-Circuit Conditions

To mimic physiological conditions, we conducted identical Ussing chamber experiments as in Figure 1 under open-circuit conditions, which allows for the investigation of paracellular ion pathways [21]. The baseline transepithelial potential difference (*V*_te_) was measured and *G*_t_ was determined from voltage deflections when applying short-current pulse. The baseline *V*_te_ was −0.9 ± 0.1 mV (*n* = 3) referenced to the serosal side. For a comparison with short-circuit conditions, equivalent *I*_sc_ was determined from *V*_te_ and *G*_t_ by applying Ohm’s law. Under open-circuit conditions (Table 1, lower rows), baseline and equivalent *I*_sc_ were not significantly different from those of short-circuit conditions. We next determined whether basal unidirectional ^22^Na^+^ flux was affected by basal *V*_te_. As shown in Table 1, *J*^Na^_MS_ and *J*^Na^_SM_ were not significantly different from those of short-circuit conditions, suggesting that baseline *V*_te_ does not affect basal transcellular and paracellular Na^+^ transport. We next measured glucose-induced transepithelial potential difference changes (Δ*V*_te_) and Δ*J*^Na^ under open-circuit conditions (Figure 3). On the one hand, addition of 10 mM luminal glucose increased the *V*_te_ (Δ*V*_te_ −6.3 ± 0.3 mV, *n* = 6). 

The equivalent ∆*I*_sc_ was 12.1 ± 0.8 µmol/cm^2^/h (Figure 3A), which was not significantly different from that of short-circuit conditions as shown in Figure 1A (*p* = 0.20). On the other hand, glucose-induced Δ*J*^Na^_MS_ increment was significantly inhibited by 60% as compared with that of short-circuit conditions (Figure 3B, Δ*J*^Na^_MS_, 4.7 ± 1.2 vs. 12.5 ± 0.8 µmol/cm^2^/h, *p* = 0.0001, open-circuit and short-circuit conditions, respectively). Interestingly, the unidirectional serosal to mucosal ^22^Na^+^ flux was significantly increased after luminal application of glucose (Figure 3B, open squares 25.2 ± 0.9 vs. 28.8 ± 1.4 µmol/cm^2^/h, *p* = 0.0003, before and after addition of glucose, respectively), which was not observed under short-circuit conditions (Figure 1B open squares). These results imply that glucose-induced luminal negativity drives the unidirectional serosal to mucosal ^22^Na^+^ flux via paracellular pathways. We next assessed the quantitative relationship between ∆*I*_sc_ and ∆*J*^Na^_Net_ under open-circuit conditions (Figure 2, open circles). There was no relationship between ∆*I*_sc_ and ∆*J*^Na^_Net_ (r square = 0.006) and the averaged Δ*J*^Na^_Net_ value (1.7 ± 1.3 µmol/cm^2^/h) was not significantly different from zero (*p* = 0.36). Taken together, these results suggest that Na^+^-dependent glucose cotransport does not concomitantly increase transepithelial Na^+^ absorption under open-circuit conditions.

### 2.4. Baseline Na^+^ Absorption Mechanisms in Claudin-15 Deficient Mice

To evaluate the impact of deficiency of claudin-15 on Na^+^ absorption in the small intestine, we first measured unidirectional ^22^Na^+^ flux across the jejunum of claudin-15 deficient (*cldn15^−/−^*) mice under short-circuit conditions (Table 2). The *J*^Na^_MS_ was decreased by 40% in *cldn15^−/−^* mice as compared with wild-type mice (31.9 ± 1.9 vs. 51.4 ± 2.3 µmol/cm^2^/h). In addition, *J*^Na^_SM_, which is mainly reflected by the paracellular pathway, was also decreased by 60% in *cldn15^−/−^* mice (10.4 ± 0.8 vs. 24.6 ± 1.7 µmol/cm^2^/h). We also observed a reduced conductance across jejunal preparations from *cldn15^−/−^* mice (17.7 ± 0.7 vs. 58.7 ± 2.2 mS/cm^2^, *p* < 0.0001 in *cldn15^−/−^* and wild-type mice, respectively). It has been shown that electrical conductance of the paracellular pathways accounts for 95% of the total conductance in the small intestine [9]. These results suggest that paracellular Na^+^-selective pores are mainly formed by claudin-15, consistent with a previous report [14]. The magnitude of the net ^22^Na^+^ flux was not significantly different than that of wild-type mice (21.4 ± 2.4 vs. 26.9 ± 1.5 µmol/cm^2^/h, in *cldn15^−/−^* and wild-type mice, respectively), suggesting that net Na^+^ absorption occurs via an electroneutral mechanism. In contrast, the basal *I*_sc_ was significantly greater than that of wild-type mice (3.3 ± 0.4 vs. 2.4 ± 0.5 µmol/cm^2^/h, in *cldn15^−/−^* and wild-type mice, respectively). Therefore, we assessed the contribution of NHE3 to ^22^Na^+^ unidirectional fluxes under baseline conditions. In the presence of the NHE3 inhibitor S3226, basal *J*^Na^_MS_ was inhibited and this magnitude of S3226-sensitive inhibition was similar to that of wild-type mice (Table 1 and Table 2, 9.6 vs. 12.5 µmol/cm^2^/h, in *cldn15^−/−^* and wild-type mice, respectively). Interestingly, the *J*^Na^_SM_ and *G*_t_ were significantly decreased by S3226 (Table 2, *p* = 0.001 and 0.018, *J*^Na^_SM_ and *G*_t_, respectively), suggesting that paracellular ion permeability was affected by S3226 in *cldn15^−/−^* but not wild-type mice. However, we did not further explore this mechanism in this study. Jointly, these results suggest that although other NHE isoforms may be involved in electroneutral Na^+^ absorption, basal net Na^+^ absorption is mostly dependent on NHE3 transport, similar to wild-type mice. In addition, paracellular Na^+^-selective pores, which are formed mainly by claudin-15, were decreased in *cldn15^−/−^* mice.

### 2.5. Na^+^-Dependent Glucose Transporter SGLT1 Is Up-Regulated in Cldn15^−/−^ Mice 

We first confirmed the expression of claudin-15 by immunofluorescence (Figure 4A) and quantitative RT-PCR (Figure 4B) experiments in the duodenum and jejunum. In wild-type mice, claudin-15 colocalized with another tight junction protein occludin. However, claudin-15 signals were completely abolished in *cldn15^−/−^* mice, consistent with a previous study [13]. It has been shown that glucose malabsorption occurs in *cldn15^−/−^* mice [14]. To elucidate the mechanism underlining these impairments, *I*_sc_ was measured in *cldn15^−/−^* mice. The addition of glucose to the mucosal side increased the *I*_sc_ in a dose-dependent manner (Figure 4C). This change in *I*_sc_ conformed to Michaelis–Menten kinetics (Figure 4D, r square = 0.996 ± 0.0001, *n* = 6), and the value of the maximum change in *I*_sc_ (*V*_max_) was three-fold increased (*p* = 0.034 by the Mann–Whitney test 1479 ± 426 vs. 426 ± 51 µA/cm^2^/h in *cldn15^−/−^* and wild-type mice, respectively).

However, the Michaelis–Menten constant (*K*_m_) was not significantly different than that of wild-type mice (*p* = 0.23 by the Mann–Whitney test, 11.4 ± 2.0 vs. 8.4 ± 2.0 mM in *cldn15^−/−^* and wild-type mice, respectively). Together, these results imply that the total number of SGLT1 transporters was increased in *cldn15^−/−^* mice to compensate for the lowered luminal Na^+^ concentration [22].

The addition of 10 mM glucose to the mucosal side resulted in an increase of *I*_sc_ (Figure 4E, ∆*I*_sc_: 23.9 ± 3.8 µmol/cm^2^/h) and *J*^Na^_MS_ (Figure 4F closed squares, 21.4 ± 2.4 µmol/cm^2^/h), while serosal to mucosal *J*^Na^_SM_ was not significantly changed after addition of luminal glucose (Figure 4F open squares, 10.0 ± 0.8 vs. 8.7 ± 0.5 µmol/cm^2^/h, *p* = 0.053, before and after addition of glucose, respectively). Since luminal Na^+^ homeostasis was disturbed in *cldn15^−/−^* mice [22], we next assessed the contribution of luminal Na^+^/H^+^ exchanger NHE3 to glucose-induced *J*^Na^_MS_ increments by using the NHE3 specific inhibitor S3226. The addition of glucose to the mucosal side resulted in an increase of *I*_sc_ (Figure 4G ∆*I*_sc_: 33.6 ± 2.5 µmol/cm^2^/h), which is slightly higher in the absence of S3226 (*p* = 0.003). In the presence of S3226, glucose-induced *J*^Na^_MS_ increment was also observed (Figure 4H closed squares, ∆*J*^Na^_MS_: 34.9 ± 2.2 µmol/cm^2^/h), suggesting that NHE3 is not responsible for glucose-induced ^22^Na^+^ unidirectional flux increments. We next assessed the quantitative relationship between ∆*I*_sc_ and ∆*J*^Na^_Net_ (Figure 5, closed circles). There was a positive correlation between ∆*I*_sc_ and ∆*J*^Na^_Net_ (r square = 0.67). Taken together, these results suggest that glucose-induced Δ*J*^Na^_MS_ is mainly mediated by SGLT1 under short-circuit conditions in *cldn15^−/−^* mice, which is the same as in wild-type mice.

### 2.6. Absence of Claudin-15 Increases Glucose-Induced Mucosal to Serosal ^22^Na^+^ Flux

We next measured basal electrical parameters and ^22^Na^+^ flux under open-circuit conditions in *cldn15^−/−^* mice (Table 2, lower rows). Baseline *V*_te_ was −6.0 ± 1.2 mV referenced to the serosal side (equivalent *I*_sc_ 4.0 ± 0.6 µmol/cm^2^/h). Under those conditions, we examined whether basal unidirectional ^22^Na^+^ flux was affected by *V*_te_. Although there was a negative luminal *V*_te_, which is preferential for increasing serosal to mucosal Na^+^ flux, the *J*^Na^_SM_ was not significantly different from those of short-circuit conditions (10.4 ± 0.8 vs. 12.0 ± 0.9 in short- and open-circuit conditions, respectively). We next measured glucose-induced *V*_te_ and *J^Na^* under open-circuit conditions (Figure 6). The addition of 10 mM luminal glucose increased the *V*_te_ (Δ*V*_te_ −20.6 ± 2.9 mV), which corresponded to an equivalent ∆*I*_sc_ (24.7 ± 2.5 µmol/cm^2^/h, Figure 6A). This equivalent ∆*I*_sc_ was not significantly different from that of short-circuit conditions, as shown in Figure 4C (*p* = 0.32). Unlike wild-type mice, there was a large negative luminal *V*_te_, and a robust glucose-induced mucosal to serosal ^22^Na^+^ flux increment was observed in *cldn15^−/−^* mice (14.5 ± 1.9 vs. 4.7 ± 1.2 mmol/cm^2^/h, *p* = 0.01, in *cldn15^−/−^* and wild-type mice, respectively, Figure 6B). After application of glucose, an increase of *J*^Na^_SM_ would be expected from such a large negative luminal *V*_te_, however, *J*^Na^_SM_ did not significantly change (Figure 6B, open squares). These results suggest that paracellular Na^+^ permeability was decreased in *cldn15^−/−^* mice. We next assessed the quantitative relationship between ∆*I*_sc_ and ∆*J*^Na^_Net_ under open-circuit conditions (Figure 5, open circles). There was no relationship between ∆*I*_sc_ and ∆*J*^Na^_Net_ (r square = 0.0012). However, the averaged Δ*J*^Na^_Net_ value (11.5 ± 2.9 µmol/cm^2^/h) was significantly different from zero (*p* = 0.01). Taken together, these results suggest that Na^+^-dependent glucose cotransport concomitantly increases transepithelial Na^+^ transport under open-circuit conditions in *cldn15^−/−^* mice.

To quantitatively evaluate the *V*_te_-dependent paracellular passive Na^+^ flux, we examined the effect of changing *V*_te_ on *J*^Na^_SM_, which mainly represents the paracellular pathway. *J*^Na^_SM_ was plotted as a function of (FΔV/RT)/{exp(FΔV/RT) − 1}, which represents the driving force of ion movement. This should yield a line having a slope of the *V*_te_-dependent diffusion flux [23]. As shown in Figure 7, the slope of the line was decreased by 50% in *cldn15^−/−^* mice as compared with wild-type mice (3.6 ± 0.9 vs. 7.8 ± 1.9, in *cldn15^−/−^* and wild-type mice, respectively), suggesting decreased paracellular Na^+^ permeability in *cldn15^−/−^* mice. Together, these results suggest that the Na^+^ which is absorbed by Na^+^-dependent glucose cotransport is recycled back into the lumen to support Na^+^-dependent glucose absorption and this Na^+^-cotransport induced luminal negative potential is important for Na^+^ recycling in wild-type mice.

### 2.7. The Efficiency of Na^+^ Recycling Systems Is Reduced in a Cholera Toxin-Induced Diarrhea Model

The preceding experiments suggest that, under physiological conditions, the Na^+^ that is absorbed by Na^+^-dependent glucose cotransport is recycled back into the lumen via paracellular Na^+^ conductance which is driven by the Na^+^ cotransport induced luminal negative potential. However, this idea is not consistent with the mechanisms of oral rehydration therapy, which is based on the notion that the Na^+^ which is absorbed by Na^+^-glucose cotransport enters the systemic circulation [16]. This discrepancy could be explained by the idea that the efficiency of Na^+^ recycling systems is reduced during infectious diarrhea. To address this directly, we measured glucose-induced unidirectional *J*^Na^_MS_ in cholera toxin-diarrhea model mice. Five hours after gavage of cholera toxin, we first verified the effect of cholera toxin on intestinal ion transport by measuring the transepithelial potential difference (*V*_te_) in isolated upper small intestine in Ussing chambers. Luminal negative *V*_te_ (referenced to the serosal side) was increased after administration of cholera toxin (–0.30 ± 0.3 vs. –2.66 ± 0.3 mV in the control and cholera toxin-diarrhea model mice, respectively, *p* = 0.008), suggesting that Cl^−^ secretion was increased in cholera toxin-diarrhea model mice. In addition, basal unidirectional *J*^Na^_MS_ was decreased by 42% as compared with the control mice (Table 3) but not for *J*^Na^_SM_, suggesting inhibition of electroneutral NaCl absorption. These observations are consistent with the action of cholera toxin on intestinal epithelial transport [24]. Interestingly, the basal *G*_t_ of cholera toxin-diarrhea model mice was significantly decreased by 27.8% as compared with the control mice (Table 3). Surprisingly, although there was such a large luminal negative *V*_te_ (2.7 mV), in which an increase of *J*^Na^_SM_ would be expected as shown in Figure 7, *J*^Na^_SM_ was actually decreased by 6.2% as compared with the control mice (Table 3). Taken together, these results suggest that paracellular Na^+^ pores, which can be formed by claudin-15, are decreased in cholera toxin-diarrhea model mice, consistent with a previous study [25]. To address this question directly, we measured *G*_t_ after stimulation of cAMP formation by forskolin and the phosphodiesterase inhibitor isobutylmethylxanthine (IBMX) in wild-type and *cldn15^−/−^* mice in Ussing chambers. *G*_t_ was decreased within 20 min after addition of forskolin and IBMX in wild-type mice (29.1 ± 1.3 vs. 21.8 ± 1.4 mS/cm^2^ before and after treatment with forskolin and IBMX, respectively, *p* = 0.0001), consistent with a previous study [26]. However, upon formation of intracellular cAMP induced by forskolin and IBMX, *G*_t_ was not decreased in *cldn15^−/−^* mice (18.6 ± 5.3 vs. 16.5 ± 5.4 mS/cm^2^, after treatment with forskolin and IBMX and control, respectively, *p* = 0.10). These results suggest that paracellular Na^+^ conductance, which is directed by claudin-15, is acutely regulated by elevation of intracellular cAMP.

We next measured glucose-induced mucosal to serosal ^22^Na^+^ flux in the cholera toxin-diarrhea model mice under open-circuit conditions (Figure 8). The addition of glucose to the mucosal side resulted in an increase in *V*_te_ (Figure 8A closed squares, Δ*V*_te_ −2.1 ± 0.3 mV, *n* = 5). As shown in Figure 8B, glucose-induced equivalent ∆*I*_sc_ was 6.7 ± 1.9 µmol/cm^2^/h, which was not significantly different from that of control conditions, as shown in Figure 3A (*p* = 0.16). Interestingly, after the addition of glucose, *G*_t_ was increased in the cholera toxin-diarrhea model mice, but not in the control mice (Figure 8C). Unlike animals that were not treated with cholera toxin treated with vehicle only (Figure 8D, open squares), robust glucose-induced mucosal to serosal ^22^Na^+^ flux increment was observed in the cholera toxin-diarrhea model mice (Figure 8D, 11.1 ± 1.8 vs. 6.1 ± 1.6 µmol/cm^2^/h, *p* = 0.08 as compared with the control conditions, in the cholera toxin-diarrhea model and the control mice, respectively). It failed to attain statistical significance, but the phenomenon is reminiscent of the *cldn15^−/−^* mice (Figure 6B). Taken together, these results suggest that the efficiency of Na^+^ recycling systems is reduced under cholera toxin diarrhea conditions.

## 3. Discussion

The aim of this study was to investigate if paracellular recirculation of Na^+^ is essential to support Na^+^-dependent nutrient absorption and to elucidate the role of Na^+^-nutrient cotransport induced luminal negative potential for Na^+^ recycling. We demonstrated that under short-circuit conditions luminal application of glucose resulted in an increment of absorptive ^22^Na^+^ fluxes (∆*J*^Na^) which corresponded to increments of short-circuit currents (∆*I*_sc_) in wild-type mice. However, under open-circuit conditions, ∆*I*_sc_ was observed but ∆*J*^Na^ was strongly inhibited. In *cldn15^−/−^* mice, a robust increment of ∆*J*^Na^ was observed under open-circuit conditions, and this recycling dysfunction was mimicked by a cholera toxin-diarrhea model in wild-type mice. Therefore, we feel that under physiological conditions, the Na^+^ that is absorbed with nutrients is recycled back into the lumen via the paracellular pathway due to pores which are formed by claudin-15. To further support this idea, the efficiency of this Na^+^ recycling system was also reduced in cholera toxin-diarrhea model mice.

### 3.1. Intestinal Nutrient Absorption Mechanisms Need a Large Amount of Luminal Na^+^

We assumed that most protein and carbohydrates are digested to monomers and absorbed via Na^+^-dependent nutrient transporters. However, it has been proposed that a significant amount of amino acids are absorbed as tripeptides by transporters driven by protons [27]. We have previously shown that the mucosal surface pH in the upper jejunum is significantly alkalinized and glycyl-sarcosine (nonhydrolyzable dipeptide, Gly-Sar) absorption was inhibited in *cldn15^−/−^* mice [22]. Furthermore, Gly-Sar induced *I*_sc_ increments were tightly coupled to luminal Na^+^/H^+^ exchange NHE3 activity and these peptide-induced *I*_sc_ increments were inhibited by NHE3 specific inhibitors S3226 and Tenapanor [22,28]. These results imply that other proton dependent cotransporters systems such as proton coupled amino acid and peptide transporters need luminal Na^+^. With respect to carbohydrates, it has been shown that complex carbohydrates can reduce the influx of carbohydrates monomers [29], which suggests that our estimation that most carbohydrates are digested to monomers is oversimplified. However, this is unlikely based on the observation that small intestinal mass absorption of glucose is mainly mediated by SGLT1, since the increase of glucose concentration in plasma after glucose gavage is reduced in SGLT1 knock-out mice [30]. Taken together, these considerations suggest that intestinal nutrient absorption mechanisms require a large amount of luminal Na^+^.

### 3.2. Paracellular Na^+^ Permeability Is Decreased in Cldn15^−/−^ Mice

It has been shown that small intestinal epithelia are classified as leaky epithelia, i.e., paracellular conductance greater than ~90% of total tissue conductance and cation selective permselectivity (P_Na_ > P_Cl_) [9]. However, the molecules responsible for permselectivity in the intestine remain to be fully elucidated. We found that electrical transepithelial conductance of *cldn15^−/−^* mice was decreased by 70% as compared with wild-type mice (Table 1 and Table 2). In addition, unidirectional ^22^Na^+^ flux from serosal to mucosal side in *cldn15^−/−^* mice, which is mainly reflected by the paracellular pathway, was decreased by 60% as compared with wild-type mice (Figure 4F and Figure 1B). Taken together, these results suggest that paracellular Na^+^ pores are mainly formed by claudin-15, consistent with a previous report [14]. Despite having defective luminal Na^+^ homeostasis [22], *cldn15^−/−^* mice do not have severe intestinal dysfunction and malabsorption (serum albumin 2.7 ± 0.19 vs. 3.0 ± 0.12g/dL, *p* = 0.17; serum total glyceride 43 ± 13 vs. 47 ± 20 mg/dL, *p* = 0.88; serum glucose 205 ± 10 vs. 270 ± 27 mg/dL, *p* = 0.1 in *cldn15^−/−^* and wild-type mice, respectively, *n* = 3 to 4 in each genotype). We believe this can be explained by the other remaining claudin(s), which could be sufficient to support the luminal Na^+^ which is needed for nutrient absorption. One possibility is that claudin-2, which forms cation-selective pores, can contribute to Na^+^ dependent nutrient absorption [11,31]. Indeed, it has been shown that claudin-2 and claudin-15 double-knockout mice die as a result of malnutrition in early infancy [32], suggesting that claudin-2 could also be contributing to Na^+^-dependent nutrient absorption.

### 3.3. Luminal Negative Potential Is Important for Na^+^ Recirculation

Our data support the conclusion that Na^+^ absorbed by Na^+^-dependent glucose cotransport is rapidly recycled back into the lumen via paracellular pathways which are driven by increased luminal negative potential generated by electrogenic glucose absorption mechanisms. Under open-circuit conditions, activation of SGLT1 did not increase mucosal to serosal ^22^Na^+^ fluxes in wild-type mice (Figure 3B). Furthermore, under the same experimental conditions, although there was a large luminal negative *V*_te_ (−20 mV), robust glucose-induced mucosal to serosal ^22^Na^+^ flux increment was observed in *cldn15^−/−^* mice (Figure 6B). To our knowledge, under physiological conditions, a postprandial robust increase in blood Na^+^ concentrations has not been shown. However, it is generally believed that there are two Na^+^ absorption systems in the small intestine; one is electrogenic nutrient-coupled Na^+^ absorption, and the other is electroneutral NaCl absorption [1]. It is also generally thought that bulk transport of NaCl absorption in the small intestine is mediated by electroneutral absorption by the coupling of luminal Na^+^/H^+^ exchanger NHE3, and Cl^−^/HCO_3_^−^ exchanger SLC26A3, since both NHE3 knockout mice and SLC26A3 knockout mice manifest in diarrhea [33,34]. These two sets of Na^+^ absorption transporter systems reside in the same nutrient absorbing enterocytes. It is predicted, therefore, that there is an interaction between the two Na^+^ absorption mechanisms. Indeed, it has been shown that Na^+^-coupled glucose uptake stimulates NHE3 transport activity in mouse jejunum [19]. However, this interaction would not be favorable for the driving force of nutrient-coupled Na^+^ absorption because a decrease of luminal Na^+^ concentration is not favorable for Na^+^-dependent nutrient absorption to absorb nutrients efficiently. Our results indicated that glucose does not stimulate NHE3 activity (Figure 1D). In addition, we fed rats with nominal Na-free diet for five days and measured intestinal luminal Na^+^ concentration. There was a significant difference in luminal Na^+^ concentrations in the stomach (59 ± 9 vs. 7 ± 1 mM in control and Na-free diet, respectively, *p* < 0.05) and colon (40 ± 8 vs. 18 ± 4 mM in control and Na-free diet, respectively, *p* < 0.05) but not in in the small intestine (57 ± 13 vs. 50 ± 9 mM in control and Na-free diet, respectively, *p* > 0.05). These results suggest that luminal Na^+^ homeostasis in the small intestine, which is the external milieu, is independent of the amount of Na^+^ intake. We believe that this luminal Na^+^ homeostasis is maintained by claudin-15 and regulated by increased luminal potential generated by electrogenic Na^+^-nutrient cotransport, since luminal Na^+^ homeostasis is disrupted in *cldn15^−/−^* mice [22]. Under pathophysiological conditions in wild-type mice, our results indicated that paracellular Na^+^ conductance was decreased by cholera toxin (Table 3). In accordance, previous studies have shown that paracellular conductance and ion selectivity were changed after treatment with theophylline (phosphodiesterase inhibitor, which raises intracellular cAMP) or cholera toxin in the rabbit ileum [25]. It has also been shown that an increase of intracellular cAMP resulted in a decrease of paracellular conductance [26,35]. Our results imply that Na^+^ conductance that is directed by claudin-15 is regulated by intracellular cAMP level. However, elucidation of molecular mechanism of this regulation requires further investigation.

### 3.4. Physiological Relevance of the Na^+^ Recirculation System in the Small Intestine

On the one hand, our conclusion implies that the Na^+^ that is absorbed by SGLT1 does not enter the systemic circulation under physiological conditions (Figure 9A). On the other hand, under pathophysiological conditions, such as in cholera-infected patients, glucose-containing oral rehydration solution (ORS) stimulates Na^+^ and water absorption, implying that the Na^+^ that is absorbed by SGLT1 does enter the systemic circulation (Figure 9B). It is also thought that glucose-induced Na^+^ absorption is not affected by cholera toxin [36]. The composition (75 mM glucose, 75 mM NaCl, and so on) of ORS is based on its efficacy in replacing water and electrolytes in individuals [16,37]. This glucose-dependent Na^+^-absorption mechanism under pathophysiological conditions is not consistent with our results under normal conditions, where Na^+^ absorbed with glucose is recycled back into the lumen rather than entering systemic circulation. Another explanation for glucose-induced Na^+^ absorption could be a decrease of Na^+^ recirculation upon infection with cholera. Indeed, our findings indicated that paracellular Na^+^ pores were decreased and glucose-induced Na^+^ absorption was observed in cholera toxin-diarrhea model mice under open-circuit conditions (Figure 8). These findings are consistent with the notion that Na^+^ recycling systems were reduced under pathophysiological diarrhea conditions. Our findings also suggest that Na^+^ cotransport-induced luminal negative potential is important for the Na^+^ recycling system. Conversely, this implies that Na^+^ absorbed via electroneutral systems, such as NaCl absorption (parallelly coupled Na^+^/H^+^ and Cl^−^/HCO_3_^−^ exchangers) and nonelectrogenic fructose absorption, can enter systemic circulation. In accordance with this notion, a recent study demonstrated that fructose-induced hypertension is initiated by increased absorption of NaCl and fructose in the intestine [38]. It is also noteworthy that ORS was occasionally associated with hypernatremia [39], implying that there is a decrease of Na^+^ recycling system activity in cases of infectious diarrhea (Figure 9B).

In summary, our data indicate that claudin-15 is important for luminal Na^+^ homeostasis and Na^+^-dependent nutrient absorption. These findings may contribute to the understanding of the mechanisms of oral rehydration therapy. Our observations raise the possibility that the Na^+^ that is absorbed with nutrients is recycled back into the lumen via paracellular pores which are formed mainly by claudin-15 under physiological conditions.

## 4. Materials and Methods 

### 4.1. Ethical Approval

All animal experimental procedures and handling were approved by the Animal Care and Use Committee of the University of Shizuoka (reference no.165117 and 175151, approved on 28th March 2016 and 8th March 2017, respectively) and conducted in accordance with the Guidelines and Regulations for the Care and Use of Experimental Animals by the University of Shizuoka.

### 4.2. Animals

Claudin-15 deficient (*Cldn15**^−/−^*) mice were originally generated in the Laboratory of Prof. Tsukita, as described previously [13]. *Cldn15**^−/−^* mice on a C57BL/6J genetic background and their age- and sex-matched wild-type mice were used. Wild-type male C57BL/6J Jcl mice from Clea Japan (Tokyo, Japan) were also used in some experiments. Mice were used at 2 to 9 months of age. The mice were fed a standard pellet diet (MF, Oriental Yeast, Tokyo, Japan), and water was provided ad libitum.

### 4.3. Measurement of Electrical Parameters and Unidirectional Fluxes of ^22^Na^+^ and ^36^Cl^−^

Mice were anaesthetized with a mixture of drugs (10 µL/g B.W., I.P. injection) consisting of medetomidine (30 µg/mL, Nippon Zenyaku Kogyo, Fukushima, Japan), midazolam (0.4 mg/mL, Teva Pharma Ltd., Nagoya, Japan), and butorphanol (0.5 mg/mL, Meiji Seika, Tokyo, Japan). The abdomen was opened by a midline incision, the small intestine from duodenum to terminal ileum was excised, and the middle one-third of the small intestine was used for experiments. The isolated segment was opened and rinsed with ice-cold oxygenized buffer to remove luminal contents, and then the muscle layer was stripped with fine forceps under a stereomicroscope. The tissue was then mounted vertically in Ussing chambers with an internal surface area of 0.2 cm^2^. The bathing solution in each chamber was 5 mL and was kept at 37 °C in a water-jacketed reservoir. The bathing solution contained (in mM) 119 NaCl, 21 NaHCO_3_, 2.4 K_2_HPO_4_, 0.6 KH_2_PO_4_, 1.2 CaCl_2_, 1.2 MgCl_2_, 0.5 glutamine, and 10 µM indomethacin, and was gassed with 95% O_2_ and 5% CO_2_ (pH 7.4). *I*_sc_ was recorded using a voltage-clamping amplifier (CEZ9100, Nihon Kohden, Tokyo, Japan). *G*_t_ was calculated from the change of current in response to voltage pulses according to Ohm’s law. We also performed experiments under open-circuit conditions to compare electrophysiological parameters and ion flux with those under short-circuited conditions. The equivalent *I*_sc_ was determined from *V*_te_ and *G*_t_ by applying Ohm’s law. The unidirectional transmural radioactive isotope fluxes of mucosal to serosal (*J*_MS_) and serosal to mucosal (*J*_SM_) were measured in adjacent tissues. Then, 9 kBq/mL ^22^Na^+^ or 1.6 kBq/mL ^36^Cl^−^ was added either to the serosal or mucosal solutions after reaching stable electrical parameters. After a 45 min period of equilibration, samples (0.5 mL each) were taken from the unlabeled side at 20 min intervals and replaced with an equal volume of unlabeled solution. Medium samples containing ^22^Na^+^ and ^36^Cl^−^ were counted in a liquid scintillation counter (LSC-7000, Aloka, Tokyo, Japan). To examine the effect of *V*_te_ change on unidirectional ^22^Na^+^ flux, we performed two 20 min flux periods with *V*_te_ at 0 mV, and two 20 min periods with *V*_te_ at 10 mV (or −10 mV).

### 4.4. Cholera Toxin-Induced Diarrhea Model

Before the administration of cholera toxin (Wako, Osaka, Japan), mice were fasted for 24 h except for water ingestion. Mice were then gavaged with a single dose of 10 µg cholera toxin in 100 µL of 150 mM NaCl solution through a gastric tube, with NaCl solution as a control. Five hours after administration, mice were anaesthetized with a mixture of drugs, the duodenum and upper part of jejunum were excised and used for the Ussing chamber experiments.

### 4.5. Chemicals

3-[2-(3-guanidino-2-methyl-3-oxopropenyl)-5-methyl-phenyl]-N-isopropylidene-2-methyl-acrylamide dihydrochloride (S3226) was synthesized by WuXi AppTec Co., Ltd. (Shanghai, People’s Republic of China). The S3226 was dissolved in 0.1% DMSO to make stock solutions. The ^36^Cl^−^ was purchased from Amersham Bioscience (Piscataway, NJ, USA), ^22^Na^+^ was purchased from Perkin-Elmer (Boston, MA, USA) and all other reagents were from Sigma (St. Louis, MO, USA).

### 4.6. Real-Time Quantitative PCR

Real-time quantitative PCR experiments were performed as previously described in [22]. The following primers were used for PCR amplifications: *Cldn15*, 5′-CAACGTGGGCAACATGGA-3′ and 5′-TGACGGCGTACCACGAGATAG-3′; *beta-actin*, 5′-CATCCGTAAAGACCTCTATGCCAAC-3′ and 5′-ATGGAGCCACCGATCCACA-3′.

### 4.7. Immunofluorescence

The small intestine was excised as in the flux experiments and opened and rinsed with ice cold PBS. The tissue segment was coated with Tissue-Tek^®^ OCT compound (Sakura Finetek, Tokyo, Japan), and embedded into a mold containing OCT compound and frozen at −80 °C. Frozen specimens were cut in 5 µm slices using a Cryostat (CM3050 S; Leica Biosystems, Nussloch, Germany) and put on coverslips. Sections were dried for 30 min, and, then, incubated in 95% ethanol on ice for 30 min. Coverslips were then bathed in acetone for one minute and rinsed 3 times in PBS. The tissue was preblocked with 5% skim milk powder in 0.1% Triton X^®^-100 in PBS (0.1% PBST) for 30 min. The coverslips were incubated with primary antibodies for claudin-15 or occludin (kindly gifted from Prof. M. Furuse, National Institute of Physiological Sciences, Okazaki, Japan) for 30 min. After washing in PBS, coverslips were incubated with secondary antibodies (1:1000 dilution) conjugated with Alexa Fluor 488 (Abcam, Cambridge, UK) or Alexa Fluor 546 (Invitrogen, Carlsbad, CA, USA). After washing, the coverslips were mounted onto glass slides with mounting medium (Fluoromount-G; SBA Southern Biotechnology Associates, Inc., Birmingham, AL, USA). Tissues were visualized using a laser scanning microscope (LSM700; Zeiss, Oberkochen, Germany).

### 4.8. Statistical and Data Analyses

Experimental values are given as the means ± SE of the indicated number of the animals. Comparisons between two groups were made with unpaired or paired Student’s *t*-test or the Mann–Whitney test. In all instances, *p* < 0.05 was considered to be statistically significant. The *K*_m_ and *V*_max_ values for the *I*_sc_ response were determined by fitting the concentration response to the Michaelis–Menten equation using nonlinear regression with GraphPad Prism software (San Diego, CA, USA).

## Figures and Tables

**Figure 1 ijms-21-00376-f001:**
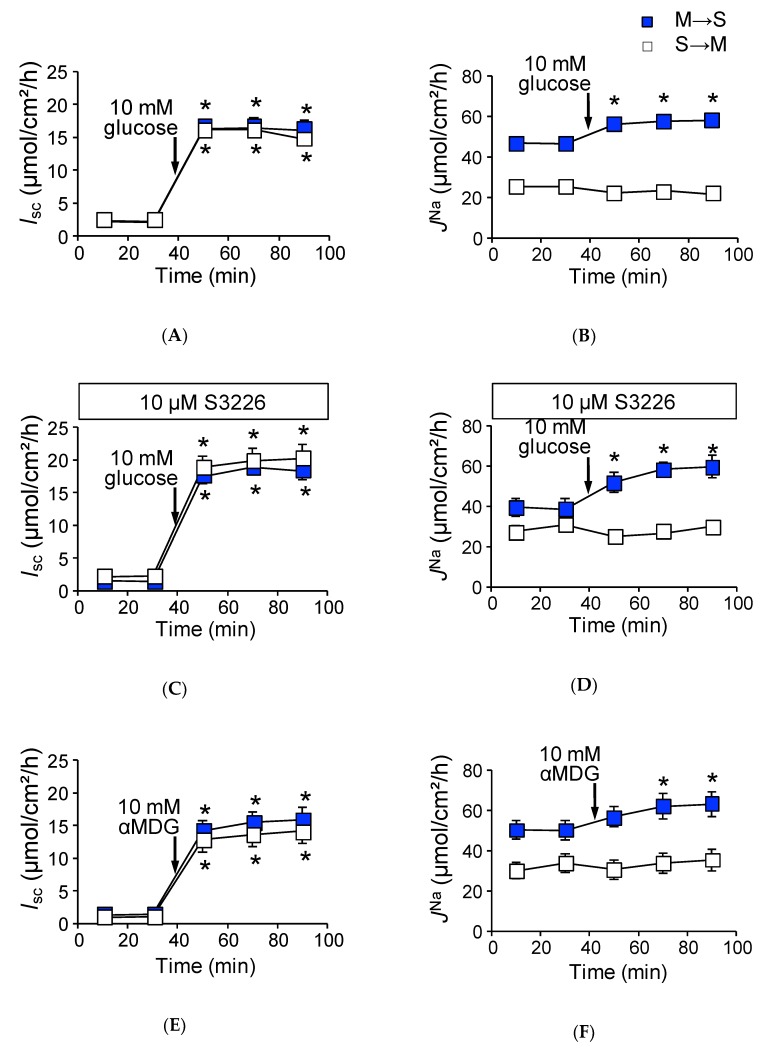
Activation of SGLT1 increases unidirectional mucosal to serosal ^22^Na^+^ fluxes under short-circuit conditions in wild-type mice: Glucose-induced short-circuit current changes (*I*_sc_) (**A**) and ^22^Na^+^ unidirectional flux changes (*J*^Na^) (**B**) were measured simultaneously in Ussing chambers, as described in the Materials and Methods. After measure basal *I*_sc_ and *J*^Na^ during the initial 30 min (squares represent mean of 9 and 8 measurements mucosal to serosal (M to S) and serosal to mucosal (S to M), respectively), glucose was added to the mucosal side, indicated by arrows (*n* = 9 and 8, M to S and S to M, respectively). The effects of S3226 on glucose-induced *I*_sc_ (**C**), and *J*^Na^ (**D**). Thirty minutes before initiation of measurement, 10 µM S3226 was added to the mucosal side (*n* = 5 and 5, M to S and S to M, respectively). Non-metabolizable sugar alpha methyl-d-glucose (αMDG) increase of *I*_sc_ (**E**) and *J*^Na^ (**F**) (*n* = 6 and 6, M to S and S to M, respectively). Closed squares indicate mucosal to serosal unidirectional ^22^Na^+^ fluxes (*J*^Na^_MS_) and open squares indicate mucosal to serosal ^22^Na^+^ fluxes (*J*^Na^_SM_). Each point represents the mean ± SE. Where error bars are absent, they are smaller than the symbol used. * *p* < 0.05 as compared with the baseline control.

**Figure 2 ijms-21-00376-f002:**
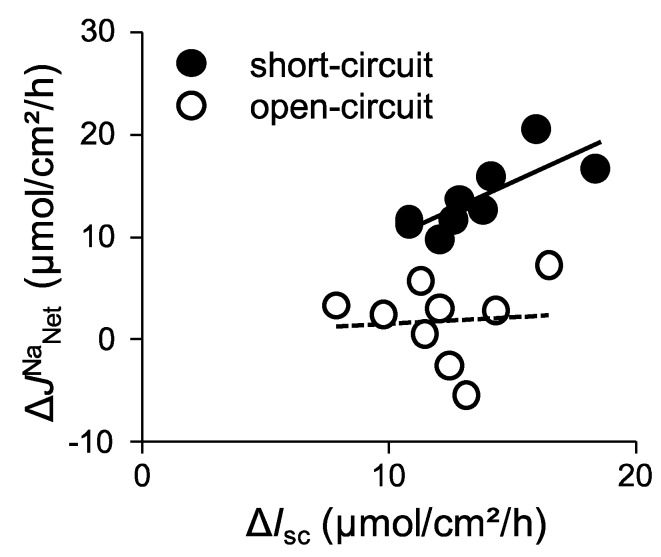
The relationship between changes of glucose-induced *I*_sc_ (Δ*I*_sc_) and changes of glucose-induced net ^22^Na^+^ fluxes (Δ*J*^Na^
_Net_) in wild-type mice: Glucose-induced Δ*I*_sc_ and net Na^+^ flux (*J*^Na^_Net_) were calculated from the data of Figure 1 (short-circuit conditions) and Figure 3 (open-circuit conditions) and replotted. Δ*I*_sc_ was determined by subtracting baseline values from those obtained after addition of glucose. Mean values for the last two periods after addition of glucose are taken as the change in *I*_sc_ and J^Na^. Net Na^+^ flux (*J*^Na^_Net_) was calculated using adjacent tissues by subtraction (*J*^Na^_MS_ − *J*^Na^_SM_ = *J*^Na^_Net_). The lines were fitted by least-squares analysis. r^2^ = 0.623 and 0.006 for the short-circuit (closed circles) and open-circuit (open circles) conditions, respectively.

**Figure 3 ijms-21-00376-f003:**
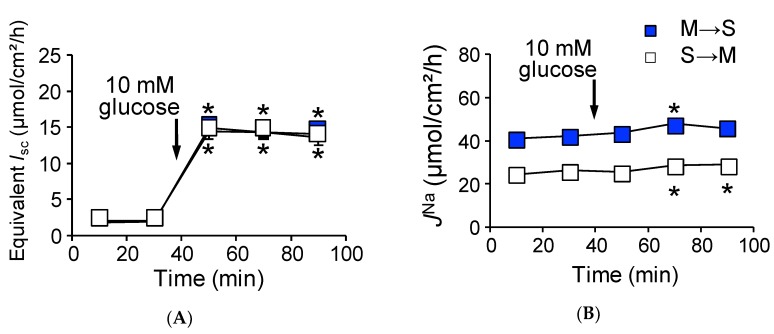
Open-circuit conditions attenuate glucose-induced *J^Na^_MS_* in wild-type mice: Glucose-induced equivalent short-circuit current changes (**A**) and ^22^Na^+^ unidirectional flux changes (*J*^Na^) (**B**) were measured simultaneously under open-circuit conditions. Equivalent short-circuit current was determined from transepithelial potential differences and transepithelial conductance by applying Ohm’s law as described in the Materials and Methods. Where indicated by the arrows, glucose was added to the mucosal side. Each point represents means ± SE (*n* = 5 and 5, M to S and S to M, respectively). Where error bars are absent, they are smaller than the symbol used. * *p* < 0.05 as compared with the control.

**Figure 4 ijms-21-00376-f004:**
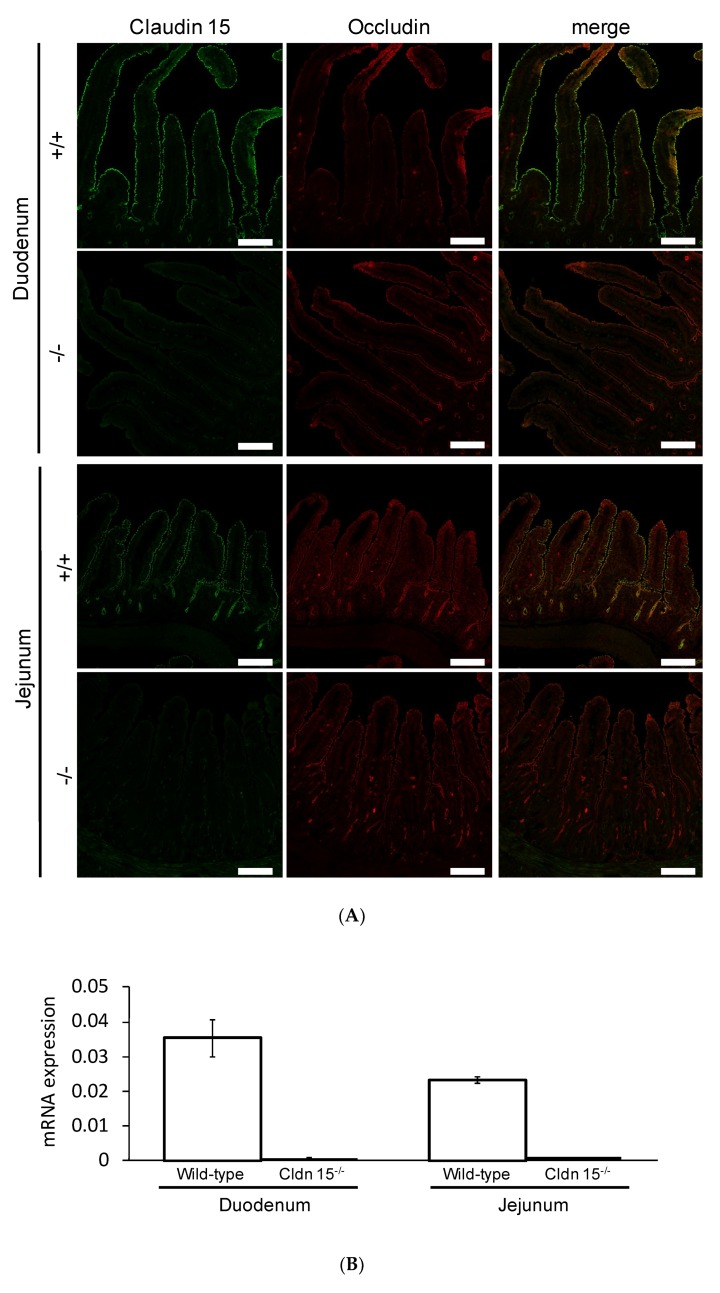
Representative confocal images of double immunofluorescence staining of claudin-15 (green) and occluding (red) (**A**) (*n* = 3) and quantitative RT-PCR (**B**) (*n* = 3) in wild-type and *cldn15^−/−^* mice. Bar, 100 µm. Activation of SGLT1 increases glucose-induced *J*^Na^_MS_ under short-circuit conditions in *cldn15^−/−^* mice: Representative *I*_sc_ trace of glucose-induced *I*_sc_ changes in *cldn15^−/−^* and wild-type mice (**C**), where, indicated by the arrows, glucose was added to the mucosal side, the final concentration of glucose is shown in mM; and the concentration dependence of the glucose-induced *I*_sc_ (**D**). The curve was fit to the Michaelis–Menten equation (*n* = 3 and 6, wild-type, and *cldn15^−/−^* mice, respectively). Where error bars are absent, they are smaller than the symbol used. The 10 mM glucose-induced short-circuit current changes (*I*_sc_) (**E**) and ^22^Na^+^ unidirectional flux changes (*J*^Na^) (**F**) were measured simultaneously the same as Figure 1 (*n* = 7 and 7, M to S and S to M, respectively). Where indicated by the arrows, glucose was added to the mucosal side. The effect of S3226 on glucose-induced Δ*I*_sc_ (**G**) and Δ*J*^Na^ (**H**) (*n* = 5 and 5, M to S and S to M, respectively). Where indicated by the arrows, glucose was added to the mucosal side. Each point represents the mean ± SE. Where error bars are absent, they are smaller than the symbol used. * *p* < 0.05 as compared with the baseline control.

**Figure 5 ijms-21-00376-f005:**
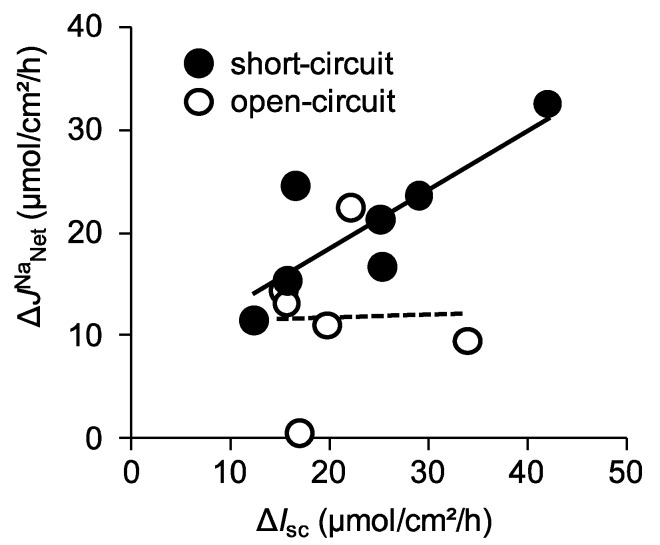
The relationship between changes of glucose-induced *I*_sc_ and changes of glucose-induced net ^22^Na^+^ fluxes in *cldn15^−/−^* mice: Glucose-induced Δ*I*_sc_ and *J*^Na^_Net_ were calculated from the data of Figure 4 and Figure 6 and replotted the same as in Figure 2. The lines were fitted by least-squares analysis. r^2^ = 0.67 and 0.0012 for the short-circuit and open-circuit conditions, respectively.

**Figure 6 ijms-21-00376-f006:**
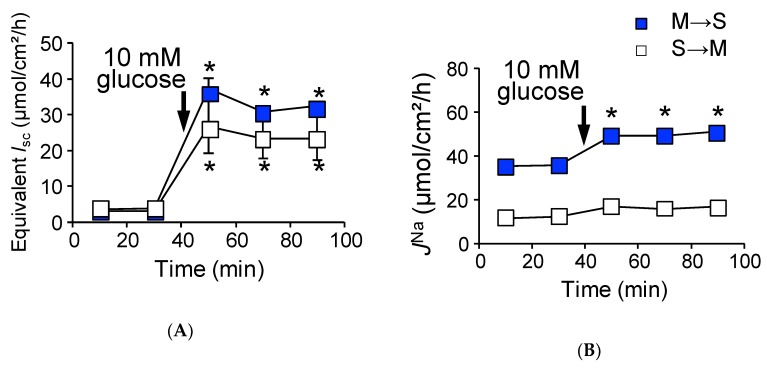
Robust glucose-induced *J*^Na^_MS_ are observed in *cldn15^−/−^* mice under open-circuit conditions: Glucose-induced equivalent *I*_sc_ (**A**) and Δ*J*^Na^ (**B**) were measured simultaneously under open-circuit conditions. Where indicated by the arrows, glucose was added to the mucosal side. Equivalent *I*_sc_ was determined the same as in Figure 3. Closed squares indicate *J*^Na^_MS_ and open squares indicate *J*^Na^_SM_ (*n* = 7 and 6, M to S and S to M, respectively). Each point represents the mean ± SE. Where error bars are absent, they are smaller than the symbol used. * *p* < 0.05 as compared with the baseline control.

**Figure 7 ijms-21-00376-f007:**
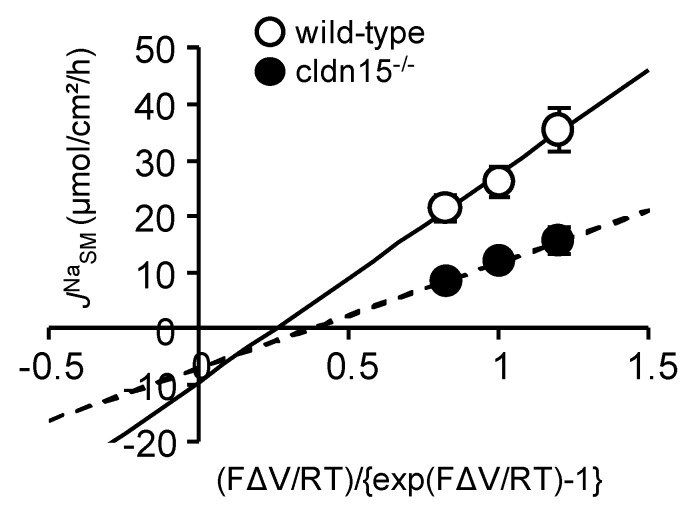
The effect of changing the transepithelial potential on *J*^Na^_SM_: *J*^Na^_SM_ was measured at *V*_te_ = 0 and ± 10 mV. *J*^Na^_SM_ was plotted as a function of (FΔV_te_/RT)/{exp(FΔV_te_/RT)−1}, where F is the Faraday constant, R is the molar gas constant, and T is the temperature. Lines are fitted with least-squares method. Wild-type (open circles, *n* = 6) and *cldn15^−/−^* mice (closed circles, *n* = 4).

**Figure 8 ijms-21-00376-f008:**
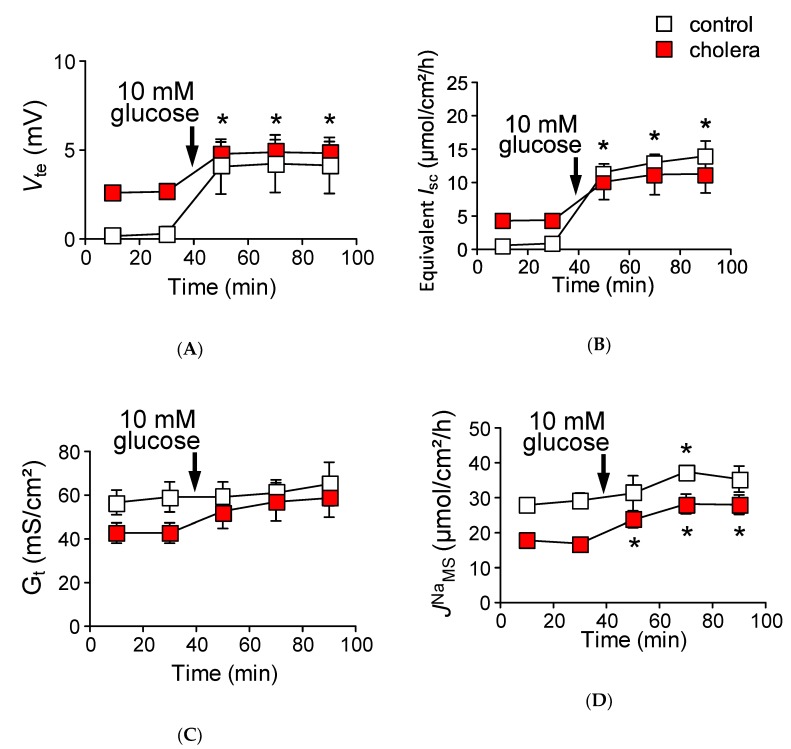
Robust glucose-induced *J*^Na^_MS_ is observed in cholera toxin-diarrhea model mice under open-circuit conditions: Glucose-induced *V*_te_ (**A**), equivalent *I*_sc_ (**B**), and Δ*J*^Na^ (**D**) were measured simultaneously under open-circuit conditions. Equivalent *I*_sc_ was determined the same as in Figure 3. *G*_t_ (**C**) was determined from transepithelial potential difference (*V*_te_) and current pulse by applying Ohm’s law. Open squares indicate control conditions (without cholera toxin) and closed squares indicate cholera toxin treated conditions (*n* = 3 and 5, control and cholera toxin treated mice, respectively). Where indicated by the arrows, glucose was added to the mucosal side. Each point represents the mean ± SE. Where error bars are absent, they are smaller than the symbol used. * *p* < 0.05 as compared with the baseline control.

**Figure 9 ijms-21-00376-f009:**
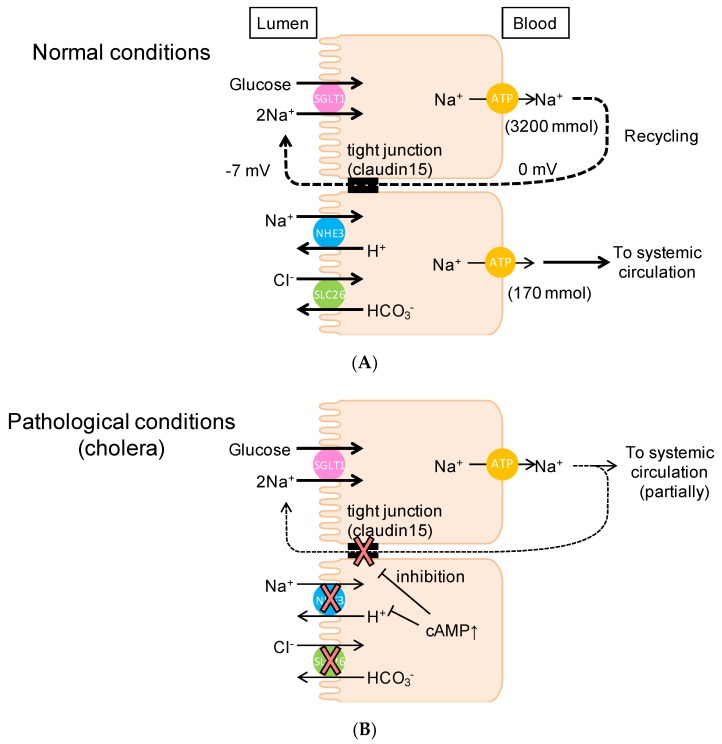
Schematic illustration of Na^+^ recycling mechanisms in the murine small intestine: Under normal physiological conditions (**A**), Na^+^, which is absorbed with glucose, is recycled back into the lumen. However, under pathological conditions (**B**), the Na^+^ that is absorbed with glucose can be partially transported to systemic circulation. Dashed lines indicate the Na^+^ recirculation and the thickness of lines indicate the amount of Na^+^. T-bars indicate inhibition. For more detail, see Discussion.

**Table 1 ijms-21-00376-t001:** Basal ^22^Na^+^ flux and electrical parameters in wild-type mice.

	*J*^Na^, µmol/cm^2^/h			*I*_sc_, µmol/cm^2^/h	*G*_t_, mS/cm^2^	*n*
	M→S	S→M	Net			
Short-Circuit Conditions						
Control	51.4 ± 2.3	24.6 ± 1.7	26.9 ± 1.5	2.4 ± 0.5	58.7±2.2	4
S3226	38.9 ± 3.4 *	28.3 ± 1.9	10.6 ± 3.9 *	1.7 ± 0.2	54.6±2.8	6
Open-Circuit Conditions						
Control	44.2 ± 2.6 ^N.S.^	22.8 ± 1.6 ^N.S.^	21.4 ± 3.7 ^N.S.^	1.8 ± 0.3 ^N.S.^	55.2±2.4 ^N.S.^	3

10 µM S3336 was added to the mucosal side. Each value represents the mean ± SE. * *p* < 0.05 as compared with control. ^N.S.^ not significant as compared with short-circuit conditions by Mann–Whitney test. M→S indicates the unidirectional mucosal to serosal Na^+^ flux. S→M indicates the unidirectional serosal to mucosal Na^+^ flux. *n*: Number of animals examined.

**Table 2 ijms-21-00376-t002:** Basal ^22^Na^+^ flux and electrical parameters in *cldn15^−/−^* mice.

	*J*^Na^, µmol/cm^2^/h			*I*_sc_, µmol/cm^2^/h	*G*_t_, mS/cm^2^	*n*
	M→S	S→M	Net			
Short-Circuit Conditions						
Control	31.9 ± 1.9 ^†^	10.4 ± 0.8 ^†^	21.4 ± 2.4	3.3 ± 0.4 ^†^	17.7 ± 0.7 ^†^	6
S3226	22.3 ± 1.5 *	5.8 ± 0.5 *	16.4 ± 1.0	3.9 ± 0.1	13.6 ± 1.0 *	4
Open-Circuit Conditions						
Control	35.6 ± 2.4 ^N.S.^	12.0 ± 0.9 ^N.S.^	23.7 ± 2.4	4.0 ± 0.6	19.4 ± 1.4	6

10 µM S3226 was added to the mucosal side. Each value represents the mean ± SE. * *p* < 0.05 as compared with the control. ^N.S.^ not significant as compared with short-circuit conditions. ^†^
*p* < 0.05 as compared with the same conditions in wild-type mice as shown in Table 1. M→S indicates the unidirectional mucosal to serosal Na^+^ flux. S→M indicates the unidirectional serosal to mucosal Na^+^ flux. *n*: Number of animals examined.

**Table 3 ijms-21-00376-t003:** Basal ^22^Na^+^ flux and electrical parameters in cholera toxin-diarrhea model.

	*J*^Na^, µmol/cm^2^/h			*I*_sc_, µmol/cm^2^/h	*G*_t_, mS/cm^2^	*n*
	M→S	S→M	Net			
Open-Circuit Conditions						
Control	29.2 ± 2.1	18.4 ± 3.6	10.8 ± 2.8	−1.0 ± 0.2	61.6 ± 9.3	3
Cholera	16.9 ± 1.4^*^	17.3 ± 1.4	−0.3 ± 1.0 *	3.9± 0.3 *	39.6 ± 1.1 *	5

Mice were gavaged without (control) or with 10 µg cholera toxin in 150 mM NaCl solution. Five hours after administration, the upper small intestine was excised and used for the Ussing chamber experiments. Each value represents the mean ± SE. * *p* < 0.05 as compared with the control by the Mann–Whitney test. M→S indicates the unidirectional mucosal to serosal Na^+^ flux. S→M indicates the unidirectional serosal to mucosal Na^+^ flux. *n*: Number of animals examined.

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
