# Peer review of "Na^+^-Coupled Nutrient Cotransport Induced Luminal Negative Potential and Claudin-15 Play an Important Role in Paracellular Na^+^ Recycling in Mouse Small Intestine"

_ijms, 2020, doi:10.3390/ijms21020376_

Round 1
Reviewer 1 Report
Nakayama et al. submitted the manuscript titled “Na+ coupled nutrient cotransport-induced luminal negative potential and claudin-15 play an important role in the paracellular Na+ recycling in the cause small intestine”, where they analyze duodenal Na+ unidirectional fluxes and short circuit currents in normal and claudin-15 knockout mice. Nakayama et al. clearly show that the application of glucose inhibits the serosal to mucosal Na+ flux, thus supporting the proposed model that serosal Na+ recicles back to drive Na+-cotransport. The manuscript is well written, presents robust results produced in well designed experiments that the authors present, interpret and discuss properly. The manuscript provides original scientific knowledge and, therefore, deserves publication int the International Journal of Molecular Sciences, after provide a new manuscript version with corrections of some minor english mistakes, the presentation of a missing table and the critical consideration of some suggestions.
Specific comments:
1. Line 4. Correct the title “…and claudin-15 plays an important role…”
2. Lines 30-31. Consider to change the sentence “…in cholera toxin-diarrhea model mice” to “…the jejunum of mice treated with cholera toxin…” to clarify and specify.
3. Line 64. Mistaken sentence, correct to “…are needed for the absorption of only one of glucose. …”
4. Line 79-81. Correct this obscure sentence. Suggestion: “… epithelia are classified as leaky epithelia, which means its paracellular conductance is ~90 % or more of the total tissue conductance and is selective to cations (PNa > PCl) [9]. …”
5. Lines 117-124. Re-writte the sentence. Suggestion: “The unidirectional mucosal-to-serosal 22Na+ flux … was larger than the serosal-to-mucosal one. …”
6. Line 124. Authors might consider to discuss why the unidirectional mucosal-to-serosal 22Na+ flux was larger than that of serodsal-to-mucosal.
7. Line 148. Obscure sentence, re-writte. Suggestion: “After measure basal Isc and JNa during the initial 30 min. (squares represent mean of X measurements), glucose was …”
8. Line 210. Missing space.
9. Line 250. Provide Table 2.
10. Discussion. The calculations described clearly in the introduction are very didactical, and have the value of strengthen the hypothesis of Na+ role in intestinal co-transport. Nevertheless, they probably are incomplete and oversimplified. Although it is valuable to keep the introduction in its actual terms, authors may consider to enrich the discussion -or the introduction- in three aspects: a) How real is the assumption that complex carbohydrates and proteins are 100 % degraded to monomers, b) It has been proposed that a significant amount of amino acids are absorbed as tri-peptides by transporters driven by protons (e. g. Smith et al. Mol Aspects Med, 34, 323-36. 2013), and c) complex carbohydrates may reduce the influx of carbohydrates monomers (e. g. Uchida et al. Int J Mol Sci 16: 10105-20).
Author Response
Dear Dr. Joanne Peng
Assistant Editor
International Journal of Molecular Sciences
Thank you for your letter regarding our manuscript ijms-675138 “Na+ coupled nutrient cotransport-induced luminal negative potential and claudin 15 plays an important role in paracellular Na+ recirculation in mouse small intestine”. We were pleased to learn that the reviewers found merit in our study. We have read the reviewers’ comments carefully and addressed all of their concerns by modifying the text according to their suggestions. We have also recognized and corrected some typo errors. The changes to the manuscript in the revised version are indicated by red-colored characters.
An itemized list of the specific changes and additions made is detailed below:
Reviewer #1
As suggested by the reviewer, we corrected the title.
Line 4. “…and claudin-15 plays an important role…”
As suggested by the reviewer, we changed the sentence.
Lines 30. “…the duodenum of mice treated with cholera toxin…”
As suggested by the reviewer, we corrected the sentence.
Line 64. “…are needed for the absorption of glucose only, …”
As suggested by the reviewer, we corrected the sentence.
Line 79-80. “… epithelia are classified as leaky epithelia, which means its paracellular conductance is ~ 90 % or more of the total tissue conductance and it is cation selective (PNa > PCl) [9]. …”
As suggested by the reviewer, we corrected the sentence.
Lines 117. “The unidirectional mucosal-to-serosal 22Na+ flux … was larger than the serosal-to-mucosal flux …”
As suggested by the reviewer, we discussed why the unidirectional mucosal-to-serosal 22Na+ flux was larger than that of serodsal-to-mucosal.
Line 124. This result suggested that Na+ absorption occurred in the baseline conditions.
As suggested by the reviewer, we corrected the sentence.
Line 149-150. “After measure basal Isc and JNa during the initial 30 min. (squares represent mean of 9 and 8 measurements M to S and S to M, respectively), glucose was …”
As suggested by the reviewer, we added space (Line 212). As suggested by the reviewer, we now provide Table 2 in Line 250. We apologize for this oversight.
Line 254. Provide Table 2.
The reviewer was concerned about our calculations. We have now explicitly stated this consideration on page 22 of the revised discussion.
Intestinal nutrient absorption mechanisms need a large amount of luminal Na+
We assumed that most protein and carbohydrates are digested to monomers and absorbed via Na+-dependent nutrient transporters. However, it has been proposed that a significant amount of amino acids are absorbed as tri-peptides by transporters driven by protons (Smith et al. Mol Aspects Med, 34, 323-36. 2013). We have previously shown that the mucosal surface pH in the upper jejunum is significantly alkalinized and glycyl-sarcosine (non-hydrolyzable dipeptide; Gly-Sar) absorption was inhibited in cldn15−/− mice (Ishizuka 2018, AJP). Furthermore, Gly-Sar induced Isc increments were tightly coupled to luminal Na+/H+ exchange NHE3 activity and these peptide-induced Isc increments were inhibited by NHE3 specific inhibitors S3226 and Tenapanor (Ishizuka 2018 AJP, Ishizuka 2019 Gastroenterology: Medicine & Research). These results imply that other proton dependent co-transporters systems such as proton coupled amino acid and peptide transporters need luminal Na+. With respect to carbohydrates, it has been shown that complex carbohydrates may reduce the influx of carbohydrates monomers (Uchida et al. Int J Mol Sci 16: 10105-20), which suggests that our estimation that most carbohydrates are digested to monomers is oversimplified. However, this is unlikely based on the observation that small intestinal mass absorption of glucose is mainly mediated by SGLT1, since the increase of glucose concentration in plasma after glucose gavage is reduced in SGLT1 knock-out mice (Diabetes 61,187-196,2012). Together, these considerations suggested that intestinal nutrient absorption mechanisms require a large amount of luminal Na+.
We believe that the manuscript is now suitable for publication in International Journal of Molecular Sciences.
Dr. Hisayoshi Hayashi
Associate Professor
Laboratory of Physiology
School of Food and Nutritional Sciences
University of Shizuoka
On behalf of all authors.
Reviewer 2 Report
The present manuscript highlights the importance of claudin 15 as well as luminal negative potential in paracellular recycling of sodium ion in murine small intestine. Authors have reported changes in unidirectional mucosal to serosal Na+ flux in WT and Cldn 15 deficient mice to investigate paracellular Na+ motion under physiological conditions. Since Na+ recycling into lumen is not consistent with the mechanism of oral rehydration therapy, authors have checked for glucose-induced unidirectional Na+ flux in cholera toxin diarrhoea mice model in order to address this disparity. The manuscript is well organized , methods are adequately described and results support the conclusion. However, there is a minor concern associated with the publication. Authors should confirm complete absence of Claudin-15 expression in the tissue of Cldn 15 deficient mice by performing western blot and qPCR analyses between WT and Cldn 15 deficient mice.
Author Response
Dear Dr. Joanne Peng
Assistant Editor
International Journal of Molecular Sciences
Thank you for your letter regarding our manuscript ijms-675138 “Na+ coupled nutrient cotransport-induced luminal negative potential and claudin 15 plays an important role in paracellular Na+ recirculation in mouse small intestine”. We were pleased to learn that the reviewers found merit in our study. We have read the reviewers’ comments carefully and addressed all of their concerns by modifying the text according to their suggestions. We have also recognized and corrected some typo errors. The changes to the manuscript in the revised version are indicated by red-colored characters.
An itemized list of the specific changes and additions made is detailed below:
Reviewer #2
We agree with you and have incorporated this suggestion into the revised discussion.
As suggested by the reviewer, we did non parametric analysis for some results.
Line 121, 307, 309, 405. “… by Mann-Whitney test.”
Line 666-667. “… or the Mann-Whitney test.”
We believe that the manuscript is now suitable for publication in International Journal of Molecular Sciences.
Dr. Hisayoshi Hayashi
Associate Professor
Laboratory of Physiology
School of Food and Nutritional Sciences
University of Shizuoka
On behalf of all authors.
Reviewer 3 Report
The paper is a good one. Experiments are well described and results seem convincing.
My primary concern is the statistical approach:
it is not possible to employ parametrical tests when the sample is tiny. In some experiments two groups of N=3 vs N=5 were compared. The authors should employ non parametric analysis.
Author Response
Dear Dr. Joanne Peng
Assistant Editor
International Journal of Molecular Sciences
Thank you for your letter regarding our manuscript ijms-675138 “Na+ coupled nutrient cotransport-induced luminal negative potential and claudin 15 plays an important role in paracellular Na+ recirculation in mouse small intestine”. We were pleased to learn that the reviewers found merit in our study. We have read the reviewers’ comments carefully and addressed all of their concerns by modifying the text according to their suggestions. We have also recognized and corrected some typo errors. The changes to the manuscript in the revised version are indicated by red-colored characters.
An itemized list of the specific changes and additions made is detailed below:
Reviewer #3
The reviewer was concerned about claudin 15 expression in cldn 15-/- mice. We performed immunofluorescence and quantitative RT-PCR experiments in the duodenum and jejunum. In wild-type mice, claudin-15 colocalized with another tight junction protein, occludin. However, claudin-15 signals were completely abolished in cldn 15-/- mice, consistent with a previous study (Gastroenterology 2008, 134, 523). We have now explicitly stated this consideration on page 13 (Fig. 4A and B) of the revised results.
We believe that the manuscript is now suitable for publication in International Journal of Molecular Sciences.
Dr. Hisayoshi Hayashi
Associate Professor
Laboratory of Physiology
School of Food and Nutritional Sciences
University of Shizuoka
On behalf of all authors.
Round 2
Reviewer 3 Report
I believe the paper is fine and may be accepted in the present form.